# Evolution and spatial reconstruction of rural settlements based on composite features of agglomeration effect and ecological effects in the Hexi Corridor, Northwest China

**Xiaoying Nie**[ORCID]*, **Chao Wang, Wanzhuang Huang**

College of Urban Construction, Lanzhou City University, Lanzhou, China

* nxy1005@163.com

## Abstract

Rural reconstruction plays a pivotal role in the revitalization of rural areas and the development of regions. Understanding the pattern and direction of rural settlement reconstruction is crucial for effectively coordinating urban and rural development, as well as promoting regional rural revitalization. The present study proposes a novel approach to elucidate the evolution and spatial reconstruction of rural settlements by integrating features of agglomeration effect and ecological effect. By employing GIS spatial analysis technology and ecosystem service value modeling, the research analyzes the combined spatial agglomeration and ecological value characteristics of rural settlements in an arid oasis area, specifically focusing on the Hexi Corridor. Based on the analysis, the study identifies specific rural settlement reconstruction zoning and directions for optimization, considering rural settlement accessibility. The study reveals three key findings: (1) There are significant differences in the scale density and spatial distribution of rural settlements across the Hexi Corridor. (2) The overall ecological environment quality is good, and there is significant spatial differentiation in the ecosystem service value, influenced by topographic factors. (3) The optimal layout mode for rural settlements in the Hexi Corridor is the combination type of 'higher-ecological higher-density'. Based on the combined agglomeration effect and ecological effect features, the research determines the reconstruction scope of alienated rural settlements. Additionally, four predominant reconstruction modes are identified: urban agglomeration type, central village construction type, internal coordination type, and ecological protection type. The study proposes viable reconstruction paths for rural settlements based on these modes.

## 1. Introduction

Since its reform and opening up, China's urbanization process has been accelerating, increasing from 17.9% to more than 60%. This large-scale urbanization process drives the flow of various production factors, such as population, land, and capital between urban and rural areas, resulting in fundamental changes to the relationship between people and land and regional functions in China's rural areas. Rural development generally faces a suite of problems, namely

**Data Availability Statement:** The data underlying the results presented in the study are available from the Data Center for Resources and

Environmental Sciences, Chinese Academy of Sciences (RESDC) (http://www.resdc.cn); The basic geographic information data set was downloaded from the Geospatial Da-ta Cloud site, Computer Network Information Center, Chinese Academy of Sciences (http://www.gscloud.cn).

**Funding:** This research was funded by the Science and Technology Program of Gansu Province (grant number 20JR10RA296). The funders had no role in study design, data collection and analysis, decision to publish, or preparation of the manuscript.

**Competing interests:** The authors have declared that no competing interests exist.

hastening non-agricultural production factors, the rapid deterioration of farmers' social subjects, increasingly empty and degraded rural construction land, serious pollution of rural water and soil environments, the dilution of kinship, and the disappearance of cultural memory symbols. In response to these problems, the central government has issued a series of strategic guidance policies to intervene in rural development. These include building a new socialist countryside, coordinating urban and rural development, targeted poverty alleviation, creating a beautiful countryside, and implementing rural revitalization. Due to that "top-down" government intervention as well as the "bottom-up" participation of market entities, the flow of urban and rural factors is becoming increasingly frequent, the speed of land transfer is accelerating, various abundant new forms of the rural economy are emerging, and vast rural areas are undergoing different degrees of reconstruction.

The spatial reconstruction of rural settlements is a key component of geographical research. Internationally, in the past 20 years such studies have mainly focused on two aspects: first is the heterogeneity of rural space and the diversification of its living subjects; second is the living experience and spatial reconstruction of different vulnerable groups who are marginalized by the 'otherization' of urban social life [1]. Relevant research results have important theoretical significance and can provide practical insight for forward-looking thinking about the trends in spatial changes and the transformation path of rural China. With the promotion of new urbanization, the rural-to-urban population flow and the reorganization and interaction of economic and social development elements are enhanced. The traditional countryside is being transformed not only by land concentration and scale management but also the adjustment of villages and towns and the industrial layout, with the industrialization, modernization, and globalization of agriculture further transforming traditional villages [2].

China's rural development has experienced unprecedented changes, and rural reconstruction has increasingly become a hot topic in academic circles. Based on different research perspectives and spatio-temporal scales, scholars have focused on the rural regional system [3];the evolution of rural settlements' spatial form [4, 5]; the differentiation and change of rural regional functions [6]; the pattern and process of rural transformation and development [7]; the mechanism, mode, and path of rural spatial reconstruction [8, 9]; the pollution and reconstruction of the rural ecological environment [10]; rurality spatial differentiation and its driving mechanism [11]; rural housing land [12, 13]; and, rural residential layout and optimization [14]. Crucially, a relatively systematic and comprehensive study can be carried out by combining typical cases with robust theoretical exploration, spatial data analysis, and social research. In terms of research methods, those based on systems science, applied statistics, GIS spatial modeling and analysis, neural networks, spatial self-organization, and empirical analysis models of relevant spatial processes have been developed [15, 16]. On the other hand, structural, humanistic, cultural value, and behavior analyses, in addition to institutional environment analysis and qualitative analytical methods have also been preliminarily applied to the study of rural spatial reconstruction [17].

In the Hexi Corridor, due to its low urbanization rate, water shortage, serious desertification, and fragile ecosystem, the rural regional system currently faces daunting environmental, ecological, and social pressures that are huge. Coupled with the rapid advancement of urbanization, the conflict between rural economic development and ecological environmental protection is particularly prominent, and rural development is now tasked with the mission of integrating limited resources and achieving rapid reconstruction [18]. The in-depth implementation of ecological civilization construction, rural revitalization, and a Western-style development strategy has brought opportunities for rural transformation and upgrading in the Hexi Corridor. Therefore, it is imperative to strengthen research addressing the reconstruction of rural social and economic forms and their spatial patterning, so as to provide a scientific basis for improving the rural spatial quality and efficient and healthy development of the Hexi Corridor.

As a result, the main research purposes were stated as follows: 1) to theoretically construct a classification system of oasis rural space from the production-living-ecological perspective; 2) to empirically explore the spatial-temporal pattern of oasis rural space; 3) to determine the reconstruction scope of alienated rural settlements and propose viable reconstruction paths for rural settlements. The rest parts of this paper were organized as follows: Section 2 introduced the research methodology and data sources, Section 3 provided the empirical results and discussion, Section 4 drew some conclusions, and then Section 5 focused on the research gaps and future prospects.

## 2. Materials and methods

### Study area

The Hexi Corridor is located in a depression zone north of the Qilian Mountains in northwest Gansu Province; it is named as such for its location west of the Yellow River and for being shaped like a corridor. Its geographical coverage is 37°17'N~42°48'N, 92°12'E~103°48'E, spanning ca. 1,000 km in length from east to west, and 100–200 km in width from north to south, with an elevation averaging about 1,500 m. The terrain of Hexi Corridor is flat, and the alluvial plain along the river forms large oases, such as those of Wuwei, Zhangye and Jiuquan, and most of its rural settlements are distributed along the inner or edge of the oases (Fig 1).

The administrative division includes five prefecture-level cities—Wuwei, Jinchang, Zhangye, Jiuquan, and Jiayuguan—and a total of 20 county-level administrative units, altogether

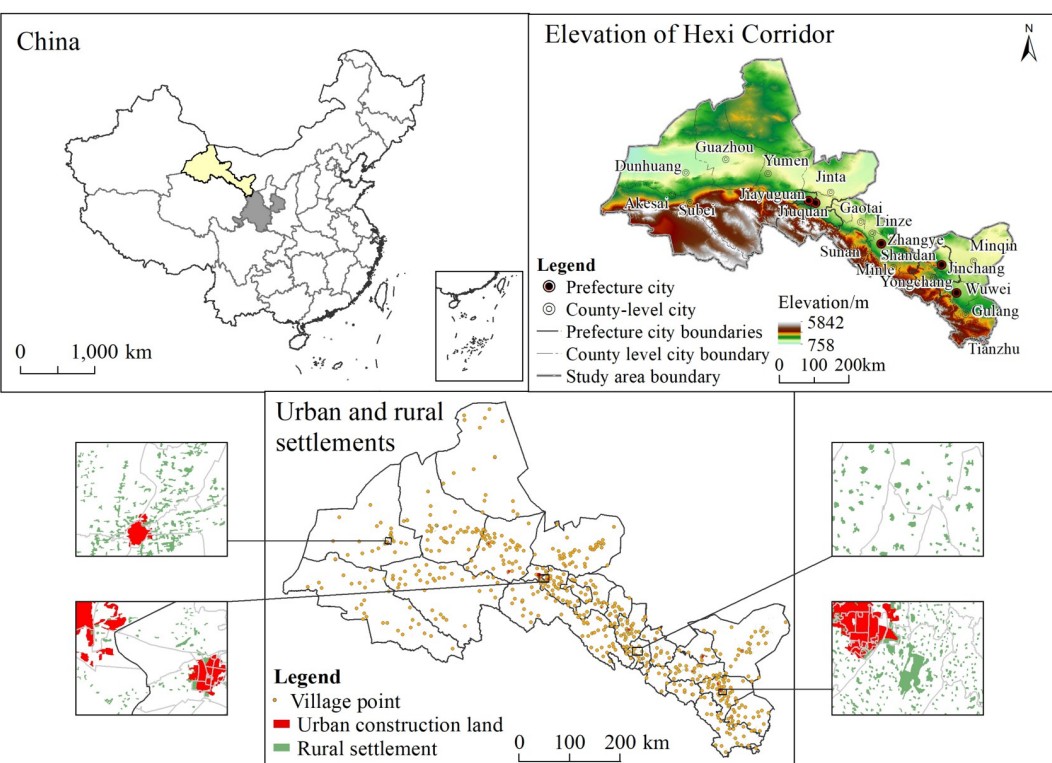

**Fig 1. Overview of the study area in Hexi Corridor, China (the map was prepared in ArcGIS 10.5 using political boundaries from the data center for resources and environmental sciences, Chinese Academy of Sciences (RESDC) (http://www.resdc.cn) and the Geospatial Data Cloud site, Computer Network Information Center, Chinese Academy of Sciences (http://www.gscloud.cn)).**

encompassing $24.57 \times 10^4$ km$^2$, this accounting for ca. 60% of the entire area of Gansu Province [19]. Hexi Corridor is a typical Gobi oasis with a long history of agricultural development and broad prospects. It serves as the main commodity grain base in Gansu Province. By 2020, the total scale of urban and rural settlements in the Hexi Corridor had reached 1093 km$^2$, of which 861 km$^2$ consisted of rural settlements.

## Data sources

The research data comprised three types: rural settlement land patch data, basic geographic information data, and statistics data. The rural settlement patch land data set was provided by the Data Center for Resources and Environmental Sciences, Chinese Academy of Sciences (RESDC) (http://www.resdc.cn). Next, as needed, land use data of the Hexi Corridor in 2000, 2010 and 2020 were extracted, accordingly. This data is generated via artificial visual interpretation based on both Landsat TM and Landsat 8 images of the United States [20]. The overall accuracy of the data is 88.95%, while the accuracy of rural settlements is at least 95% [21].

The basic geographic information data set was downloaded from the Geospatial Data Cloud site, Computer Network Information Center, Chinese Academy of Sciences (http://www.gscloud.cn). This includes information about the administrative division boundaries, transportation networks, the river system, and digital elevation model (DEM), etc.; the DEM data has a 30 m spatial resolution.

The grain output data used to calculate the service value per unit area of each ecosystem came from the *Gansu Development Yearbook* (2009, 2011, and 2021); the grain price was taken from the National Agricultural Product Cost and Income Data Summary 2021. The CPI index was downloaded from the website of the National Bureau of Statistics in China. ArcGIS 10.5 software was chiefly used for data processing and analysis.

## Methods

**Classification system of "production-living-ecological space".** The identification and classification of "production-living-ecological space" is the basis for the study of spatio-temporal evolution characteristics of rural settlements [22]. Drawing on the identification schemes of "production-living-ecological space" in the literature [23–26], this paper applied the classification system based on remote sensing monitoring data of land use in China, according to the spatial meaning [27] and the actual situation of land use in the Hexi Corridor. Then, guided by highlighting the diversity and subjectivity of land functions [28], the subcategories of the existing system were reclassified accordingly, as detailed in Table 1.

**GIS spatial analysis.** The average nearest neighbor index (ANN) describes the spatial distribution characteristics of point elements assuming the distribution of random patterns. In this paper, the ANN was used to infer and judge the spatial distribution pattern of rural settlements in Hexi Corridor. By calculating the average distance between the center point of a given rural settlement patch and that of the nearest settlement, and then comparing it to the expected average distance assumed under a random distribution, the agglomeration and dispersion characteristics between urban and rural settlements can be determined accordingly. The calculation formula is as follows [29]:

$$ANN = \frac{\overline{D_0}}{\overline{D_e}} = \frac{\sum_{i=1}^{n} {d_i}/{n}}{\sqrt{{A}/{(2n)}}} \tag{1}$$

where, $\overline{D_0}$ is the observed average distance value; $\overline{D_e}$ is the expected mean distance value; $n$ is

Table 1. Classification system of "production-living-ecological space".

| First-level | Corresponding land use type | Classification standard |
|---|---|---|
| Ecological space | Forest land | Refers to the forestry land where trees, shrubs, and other natural and artificial forest vegetation is grown, and the canopy density is at least 10%: woodland, shrubland, and sparse woodland (3 subcategories). |
| | Grassland | Refers to all kinds of grassland dominated by herbaceous plants with a coverage > 5%, including high-coverage grassland, medium-coverage grassland and low-coverage grassland. |
| | Water areas | Refers to natural land areas and land for water conservancy facilities, having 5 categories: rivers and canals, lakes, reservoirs, ponds, permanent glaciers and snow, and beaches. |
| | Unused land | Refers to land not yet currently used: namely, sandy land, Gobi land, saline alkali land, marshland, bare land, bare rock and rocky land, and others. |
| Production-ecological space | Arable land | Arable land which includes paddy fields and dry land. |
| | Garden land | Refers to unforested land, nursery sites, and all kinds of garden/horticultural land. |
| Living-production space | Rural settlements | Refers to rural settlements that are independent of towns and cities. |
| | Urban industrial and mining land | Refers to the land of large, medium, and small cities and built-up areas above counties and towns, plus any other land that is not industrial, mining, or transportation land. |

the total number of rural settlement patches; $d$ is the distance; $A$ is the study area. If the *ANN* value is less than 1, the pattern is one distinguished by clustering; if the equal to 1, the pattern is uniformly distributed; and, if greater than 1, the pattern is one that tends to be diffuse.

Local spatial autocorrelation (Getis-Ord Gi*) index can be used to evaluate the degree of spatial agglomeration in the rural settlement distribution across the study area. Further, it can reveal spatially localized cold and hot spots in that distribution's range. In this paper, the Gi* index was used to measure the spatial auto-correlation of rural settlements, with their spatial distribution characteristics in the study area and their impact on the neighborhoods also evaluated [30]. The calculation formula is as follows:

$$G_i^* = \sum_{j=1}^{n} w_{ij} x_j / \sum_{j=1}^{n} x_j \tag{2}$$

Where, $w_{ij}$ denotes the spatial weighting, as defined by the distance rule; $x_i$ and $x_j$ denote the rural settlement patch areas of region $i$ and $j$, respectively.

**Trend surface analysis.** Trend surface analysis is a way to simulate the distribution law and changing trend of geographical elements in space by using a mathematical surface. Based on the number and area data of rural settlements at three time points over a 20-year period (2000, 2010, and 2020), we analyzed the spatial differentiation characteristics of rural settlements in the Hexi Corridor. Let $Z_i(x_i, y_i)$ be the actual observed value of the $i$ geographic element, let $(x_i, y_i)$ be the coordinate value of plane space, for which axis $X$ represents the east–west direction while axis $Y$ represents the north–south direction.

**Kernel density estimation.** Kernel density estimation (KDE) is the most widely used non-parametric estimation technique among the methods for spatial point model analysis [31], because it can directly convey the spatial distribution pattern and evolution trend of rural settlements [32]. In nuclear density mapping, the density contribution values of points falling into each grid cell within a specified range (i.e., a circle with radius $h$) follows a distance attenuation effect. The value of the points represents the degree of rural settlement agglomeration in the spatial distribution. The calculation formula is as follows:

$$f(x) = \frac{1}{nh} \sum_{i=1}^{n} k\left(\frac{x - x_i}{h}\right) \tag{3}$$

where, $k\left(\frac{x-x_i}{h}\right)$ is the kernel function; $n$ is the number of rural settlements on the landscape; $h$ is the bandwidth, and $h > 0$; $(x - x_i)$ is the distance from the estimated point $x$ to the sample point $x_i$.

**Ecosystem services value evaluation model.** The principles and methods of ecosystem services value estimation were first introduced and carried out by Costanza et al. [33], to quantify ecosystem service values on a global scale. Later, Xie et al. [34] further clarified the ecological value information corresponding to differing land use types based on China's reality. This method can better represent the actual ecological environment of rural settlements, and can thus provide support for the reconstruction of rural spaces from the perspective of ecological environment protection. The calculation model for this is as follows:

$$ESV = \sum_{k=1}^{n} (A_i \times VC_i) \tag{4}$$

where, $ESV$ is the total annual value of ecosystem services in the study area; $A_i$ is the area of the class $i$ land use type; and, $VC_i$ is the ecosystem service value coefficient of the class $i$ land use type.

Here, the $ESV$ per unit area scale, as revised by Xie et al. [34] in 2015, was adopted to evaluate the ecological service value of each ecosystem type in the Hexi Corridor. The value equivalents of 'forest land', 'grassland', 'water area', and 'arable land' correspond to forest, grassland, water area, and farmland ecosystems, respectively. The value equivalent of 'unused land' is the sum of wetland and desert ecosystems. The ecological service value of 'garden land' is between arable land and forest land, namely, the average value for farmland and forest ecosystems. The 'rural settlements' correspond to desert ecosystems, and 'urban industrial and mining land' corresponds to the sum of desert and bare land ecosystems. Those value equivalents were calculated according to Liu's research [35]. On this basis, by referring to the relevant work of Xie et al. [36], we set 0.42 as the correction coefficient for the value equivalents of the Hexi Corridor to obtain its ecosystem services value equivalent factor (Table 2).

According to the value of one equivalent factor being 1/7 of the national average market price of grain per unit yield [37], most studies have set grain price as the average of a certain year, or several years, when studying the evolution of ecosystem services value. Since the change in ecosystem services value has a time-dependent effect, following several pertinent studies [38, 39], we derived the grain prices of corresponding years, respectively. Further, we used the national grain-fixed CPI data to eliminate the impact of price fluctuations caused by inflation, to obtain non-biased values of unit equivalent factors in 2000, 2010, and 2020 (Table 3).

**Table 2. Equivalent factors of ecological service value.**

| First-level | Corresponding land use type | Ecosystem services value equivalent factor |
|---|---|---|
| **Ecological space** | Forest land | 33.09 |
| | Grassland | 15.20 |
| | Water areas | 57.07 |
| | Unused land | 22.39 |
| **Production-ecological space** | Arable land | 3.32 |
| | Garden land | 18.21 |
| **Living-production space** | Rural settlements | 0.37 |
| | Urban industrial and mining land | −0.18 |

Table 3. Correction results for the unit equivalent factor values in the Hexi Corridor.

| Year | National grain price (yuan/kg) | Grain output per unit area (kg/hm²) | Unit output value (yuan /hm²) | Unit equivalent factor value (yuan) | National grain CPI (year 2000 = 100) | Corrected unit equivalent factor value (yuan) |
|---|---|---|---|---|---|---|
| **2000** | 0.97 | 2549.77 | 2473.28 | 353.33 | 100.00 | 837.14 |
| **2010** | 2.08 | 3422.77 | 7119.36 | 1017.05 | 176.50 | 1364.88 |
| **2020** | 2.45 | 4556.78 | 11164.11 | 1594.87 | 236.93 | 1594.87 |

**Measuring the accessibility of rural settlements.** Considering the daily travel habits and travel conveniences of rural residents, the minimum time spent by rural settlements to reach the destination was used here for the quantitative analysis of accessibility; smaller the value is, the better the accessibility is [40]. According to the combined characteristics of agglomeration and ecological value, the high ecological density type area was taken as the study area's center, in which rural settlements have good traffic conditions, ecological environment, and reconstruction advantages. The accessibility range of rural settlements was measured using the Cost Weighted Distance tool in ArcGIS. When the path distance is used to measure the spatial accessibility, the accessibility is related to the start point, the end point, and the land use type between these two points. Under different land use types, however, the time cost of passing each unit of distance differs. Considering that the main travel mode of villagers to reach the urban industrial and mining land is either private car or bus, the main way to reach the village center, arable land or other land types is by walking there. Following relevant studies [41, 42], the traffic speed of various land use types was converted into the traffic time cost of grid cells. Based on the topographic characteristics of the Hexi Corridor, this study also considers the influence of slope upon traffic dynamics (Table 4).

The key steps in quantifying spatial accessibility are as follows. (1) Size dimensions of raster data are set to 30 m × 30 m, and the cost is defined as the degree of elapsed time required to traverse each land use type's raster data (i.e., time cost). (2) Take the average time (min) needed to travel 30 m as the value for that time cost. (3) Use ArcGIS to assign values to the layers of each land use type, and then obtain the minimum time cost raster after transformation and spatial superposition; finally perform the accessibility measurement.

## 3. Results and discussion

### Spatial and temporal differentiation of rural settlements

Rural settlement patches varied greatly in area and their scale is generally small. In 2000, the maximum patch area of a rural settlement was 3.30 km², and the average patch area was 0.08

Table 4. Minimum time cost of the nine major land use types in the Hexi Corridor.

| Land use type | Speed (km/h) | Minimum time cost (min) |
|---|---|---|
| **Urban industrial and mining land** | 60 | 0.03 |
| **Rural settlements** | 4 | 0.45 |
| **Arable land** | 3 | 0.60 |
| **Garden land** | 3 | 0.60 |
| **Forest land** | 3 | 0.60 |
| **Grassland** | 3 | 0.60 |
| **Water areas** | 3 | 0.60 |
| **Unused land** | 3 | 0.60 |
| **Slope > 25˚** | 1 | 1.80 |

km² ; in 2020, corresponding values were 9.29 km² and 0.10 km² . Using the ANN method, we obtained 0.28, 0.29 and 0.30 for the statistical values of R in 2000, 2010 and 2020, with standardized Z-values of –130.32, –127.56, and –122.47, respectively; importantly, all P values were $\ll$ 0.0001. Taken together, these results showed that the agglomeration trend of rural settlement spatial distribution pattern in Hexi Corridor in 2000, 2010, and 2020 was significant and intensified over time.

The trend surface analysis model was then used to analyze the number and area of rural settlements within township units, to further explore the overall change trend of rural settlements in the Hexi Corridor. The number and area of rural settlements per township unit were taken as the height attribute value (Z-value), and a 3D perspective was generated via fitting (Fig 2). Evidently, along the east–west direction, the number of settlements is significantly higher in the east and lower in the west, but vice versa for settlement area. This revealed that rural settlements in the Hexi Corridor are mainly distributed in a low-density scattered pattern in the west and a high-density cluster pattern in the east.

The number of rural settlements along the north–south direction shows an inverted U-shape, that is, the spatial distribution pattern is more pronounced in the middle and less so in the north or south; as before, the settlement area also shows the opposite pattern. The reason being that the Hexi Corridor's central part is flat and thus the main distribution area of oases. Further, the Lanzhou-Xinjiang Railway and G312 run through the whole territory, which draws more rural settlements to gather there. Collectively, this shows a unique law of an arid area

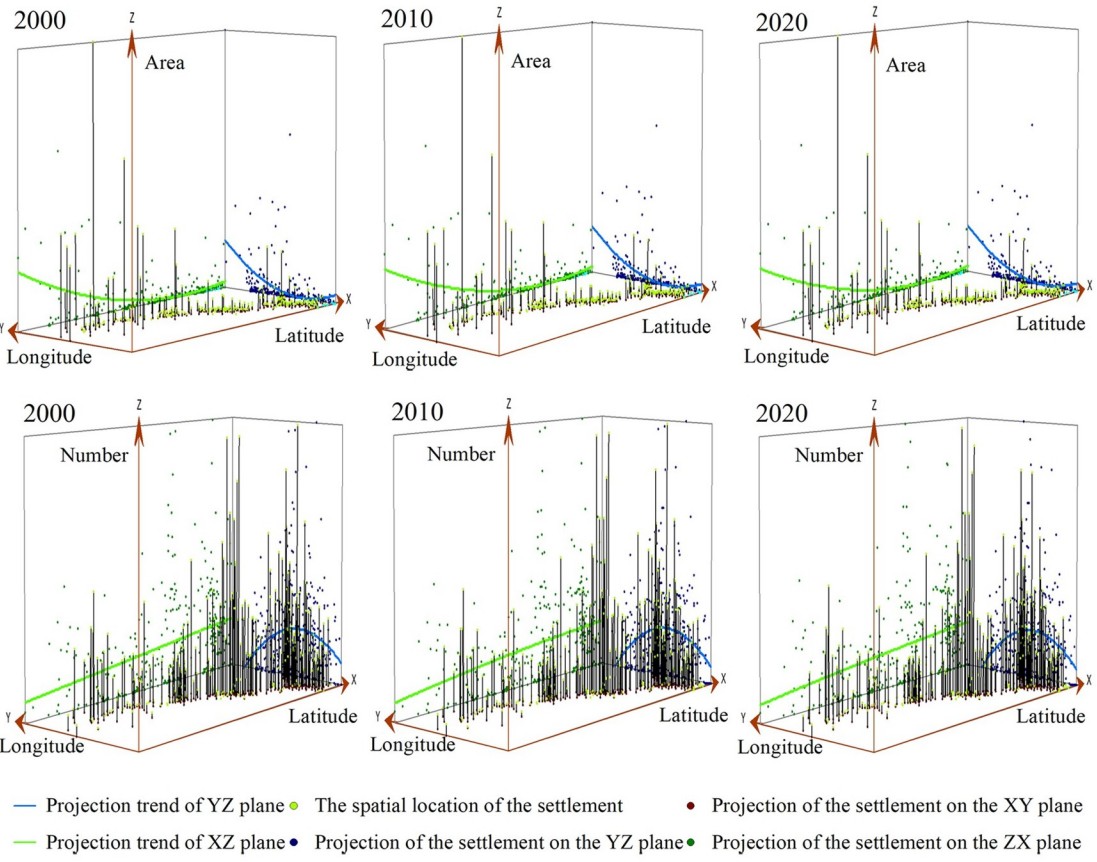

Fig 2. Trends in the number and area of rural settlements in the Hexi Corridor.

being dominated by low-density large-scale agglomeration distribution at its north and south ends and high-density small-scale agglomeration distribution in its middle. Viewed temporally, in contrast, the overall spatial trends in the number and area of rural settlements is not obvious.

According to the kernel density of rural settlements (Fig 3), it can be seen that, firstly, the density of rural settlements in the Hexi Corridor has decreased as a whole from 2000 to 2020. Secondly, the spatial distribution of rural settlement density was significantly different, forming two core density zones in the middle and east of the corridor, which are clustered and distributed along rivers, traffic trunk lines, towns and oases. Thirdly, the density value showed spatially gradient attenuation, being higher in the center but lower towards the periphery. It is roughly consistent with the geomorphic pattern of "two mountains and three basins" in the Hexi Corridor. We detected two density core zones of rural settlements in the Hexi Corridor, whose kernel density range was 0.68–2.84/km$^2$, which includes the plain areas of Wuwei, Jinchang, and Zhangye. The terrain is relatively flat, water resources are relatively sufficient, and the cultivated land conditions are good, all of which is conducive to the development and development of rural settlements.

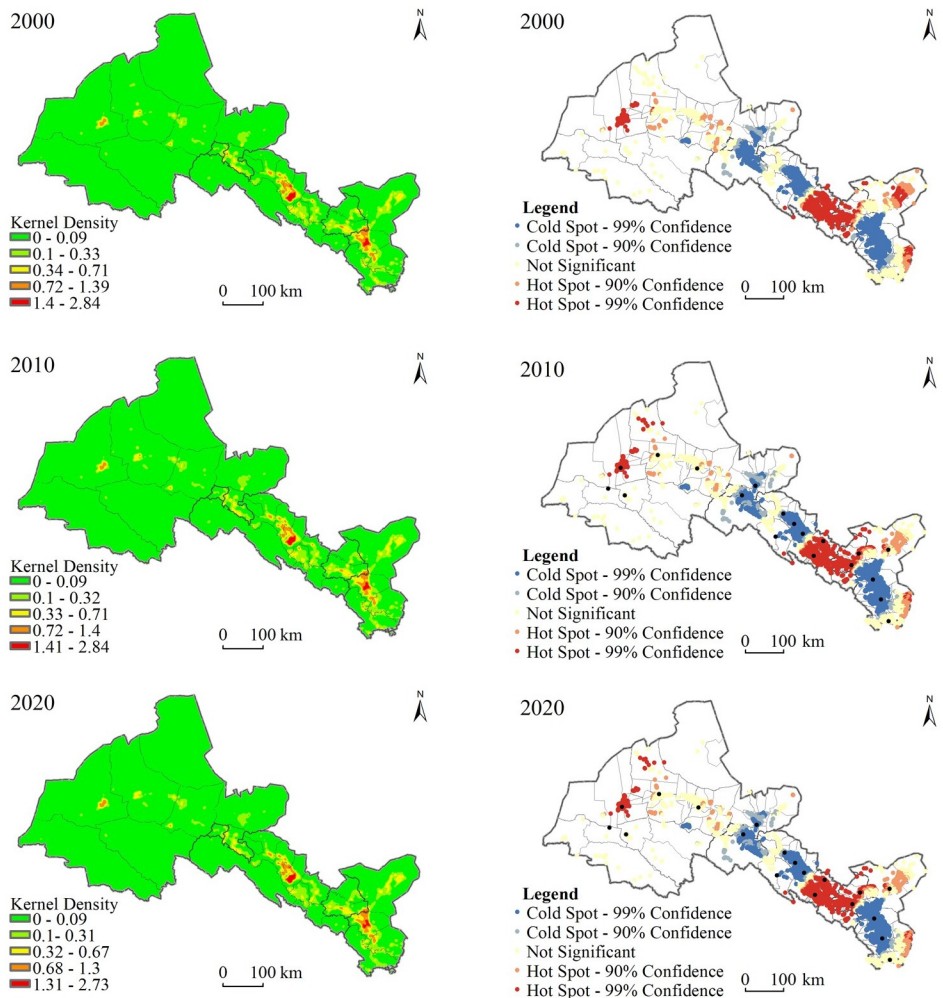

**Fig 3. Spatial distribution of the scale core density value and the hot and cold spots of rural settlements in the Hexi Corridor (the map was prepared in ArcGIS 10.5 using political boundaries from the data center for resources and environmental sciences, Chinese academy of sciences (RESDC) (http://www.resdc.cn)).**

The global spatial clustering test was used to analyze the scale characteristics of rural settlements. The resulting Z-values of 2000, 2010, and 2020 were –10.58, –10.38, and –6.34, respectively, indicating that rural settlements in Hexi Corridor presented a global small-scale agglomeration feature. The spatial local hot spot detection method was used to measure the spatial differentiation characteristics of cold vs. hot spots of the rural settlement scale, and a corresponding map of rural settlement scale distribution was drawn with the settlement patch area as the statistical attribute (Fig 3). These results showed that from 2000 to 2020, rural settlement hot spots in the Hexi Corridor were mainly concentrated in Jinchuan, Yongchang, Shandan, Minle, Dunhuang, and Minqin. After 2010, however, most of the rural settlement hotspots with high value in Minqin turned into sub-hotspot areas, which we attributed to ecological migration caused by ecological environment problems in Minqin and the reduced rural settlement scale. Rural settlement cold spots were distributed in Liangzhou, Gulang, Ganzhou, Linze, Gaotai, Sunan, Suzhou, and Jinta, thus indicating that the scale of villages in those regions was small. After 2010, some of the low-value clustered cold spots in Suzhou became sub-cold spots, while the cold spots in other regions did not change significantly. The reason is that in the rural settlement scale system of Suzhou District, many small settlement patches expanded rapidly, while in other regions the agglomeration effect of large-scale settlement patches was slight or negligible; hence, numerous small settlement patches did not undergo integration. This suggested the spatial agglomeration degree of rural settlement land to urban areas and central towns was not prominent.

## Temporal and spatial distribution characteristics of ecological values

According to formula (4), the ecosystem services value of each spatial type under the "production-living-ecological space" system in the study area was calculated (Table 5). From 2000 to 2020, the ecosystem services value of Hexi Corridor increased by 89%, from 420.797 to 795.301 billion yuan, an increase of 374.504 billion yuan. Comparing the two decades, the 62.26% increase during 2000–2010 surpassed the slower increase of 16.48% in 2010–2020. Notably, ecological space contributed the most to the total ecosystem services value, in particular the unused land occupying the largest area of the Hexi Corridor; it played a pivotal role, with a contribution rate of 74.93%, indicating that the overall ecological environment quality in the study area was good. The pooled contribution rate of production-ecological space was about 1.03%, while that of life-production space to ecosystem services value was the least (0.01%). Over time, the ecosystem services value of urban industrial and mining land decreased, while

**Table 5. Changes in the ecosystem services value (ESV) of the "production-living-ecological space" in the Hexi Corridor, from 2000 to 2020.**

| Production-living-ecological space | Land-use type | ESV (×10$^8$ yuan) | | | | Percentage change (%) | Average contribution (%) |
|---|---|---|---|---|---|---|---|
| | | **2000** | **2010** | **2020** | **Absolute change** | | |
| **Ecological space** | Forest land | 202.89 | 329.02 | 384.31 | 181.42 | 89.42 | 4.82 |
| | Grassland | 679.57 | 1108.09 | 1291.79 | 612.22 | 90.09 | 16.21 |
| | Water areas | 124.69 | 201.39 | 245.07 | 120.37 | 96.53 | 3.00 |
| | Unused land | 3159.72 | 5117.45 | 5945.34 | 2785.62 | 88.16 | 74.93 |
| **Production-ecological space** | Arable land | 39.53 | 69.45 | 83.59 | 44.06 | 111.48 | 1.00 |
| | Garden land | 1.39 | 2.32 | 2.66 | 1.27 | 91.12 | 0.03 |
| **Living-production space** | Rural settlements | 0.24 | 0.40 | 0.51 | 0.27 | 110.86 | 0.01 |
| | Urban industrial and mining land | –0.06 | –0.11 | –0.26 | –0.20 | –344.37 | 0.00 |
| **Total** | | 4207.97 | 6828.01 | 7953.01 | 3745.04 | 89.00 | 100 |

the ecosystem services value of other land types increased, among which the cultivated land and rural residential area rose the fastest, with an average annual growth rate of 5.56% and 5.54%, respectively. Unused land development and various immigration projects are the main reasons for the changed ecosystem services value of cultivated land and rural residential areas.

We further studied the spatial characteristics of ecosystem services value in the Hexi Corridor, by considering the influence on it from human activities in the oases up, combined with the cultivation radius and activity radiation range of settlements, and by referring to relevant research [43]. Thus, the study area was divided into a grid consisting of 3km × 3km units. Spatial differences in the ESV in the Hexi Corridor in 2000, 2010, and 2020 were respectively obtained, and these then divided into five levels according to the natural break point method (Fig 4). As seen in the figure, the ecosystem services value of the Hexi Corridor displayed significant spatial differences. The high-value area is mainly distributed in the southern Qilian Mountains and the northern desert portion, while the low-value area is concentrated in the central oasis plain. That pattern is the opposite found for rural settlement concentration degree, which indicated that the spatial distribution of ecosystem services value is clearly affected by topographic factors.

From 2000 to 2020, the areas with higher values were basically stable, being mainly distributed in the Qilian Mountains and Heli Mountains where there is complex terrain, rich forest resources, and a good ecological environment quality. There, rural settlements were evidently scattered and their degree of agglomeration was weak. For the lower ecological value, its distribution was mainly concentrated in the oasis plain area running along the middle of the

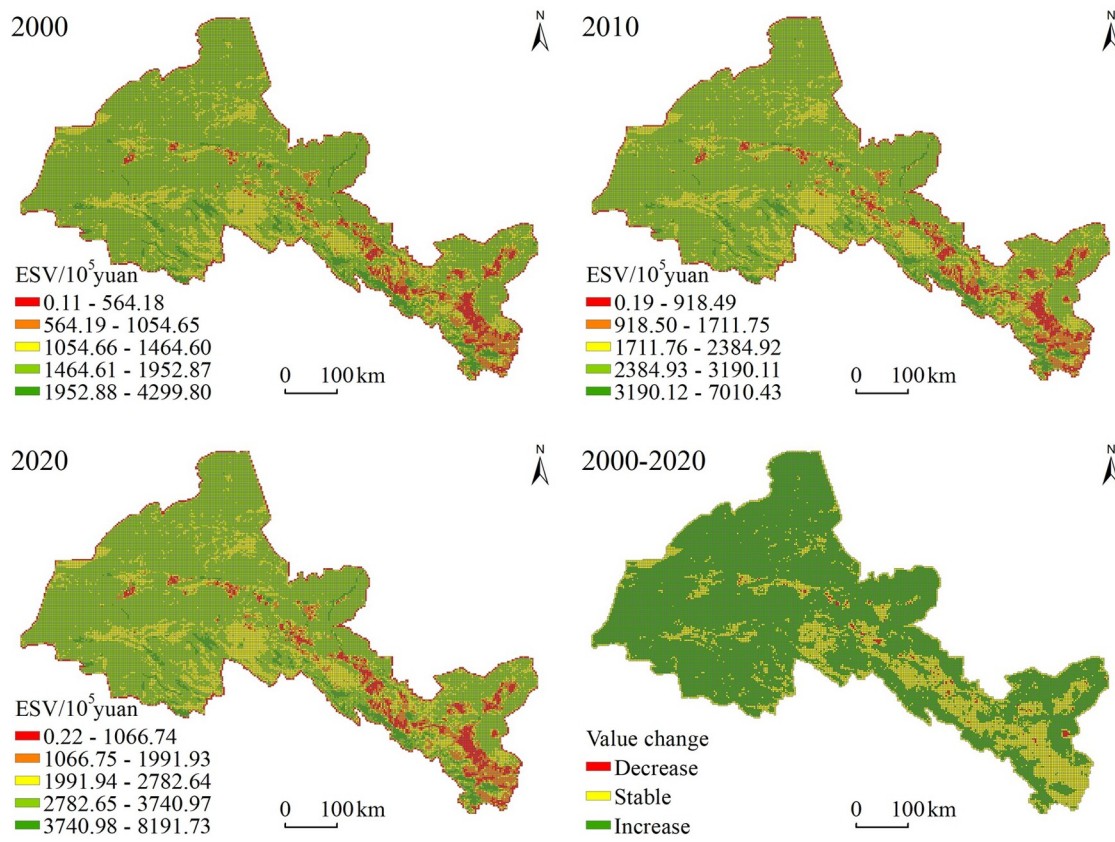

**Fig 4. Temporal and spatial changes in the ecosystem services value (ESV) across the Hexi Corridor(the map was prepared in ArcGIS 10.5 using political boundaries from the Data Center for Resources and Environmental Sciences, Chinese Academy of Sciences (RESDC) (http://www.resdc.cn)).**

corridor. There, being affected by urban land expansion, the rural settlement concentration degree is high, and the ecological environment quality is conversely relatively poor.

Nevertheless, the spatial pattern evolution characteristics of ecosystem services value showed that on an area-basis, it was increasing more than decreasing over. The central oasis area was greatly affected by human activities such as urbanization, constituting the main zone where ecosystem services value declined. The mountain and desert areas in the north and south of the Hexi Corridor were less affected by human activities, and ecological protection measures implemented there in recent years have augmented ecosystems service value in those two regions. The stable area of ecosystem services value was concentrated in the fringe area of oases. In this way, the spatial pattern and evolution characteristics of ecosystem services value revealed, to some extent, the future development potential and direction of rural settlements.

## Composite features of agglomeration effect and ecological effects

ArcGIS was used to superposition the core density map of rural settlements and the spatial distribution map of ecological value, to identify the combined characteristics of rural agglomeration and ecological value distribution. This yielded a total of five types of combination areas (Fig 5, Table 6). Of these combination types, the high-ecological high-density one is the optimal layout mode, because it can realize the coordinated development of regional ecological environment protection and intensive and economical use of land.

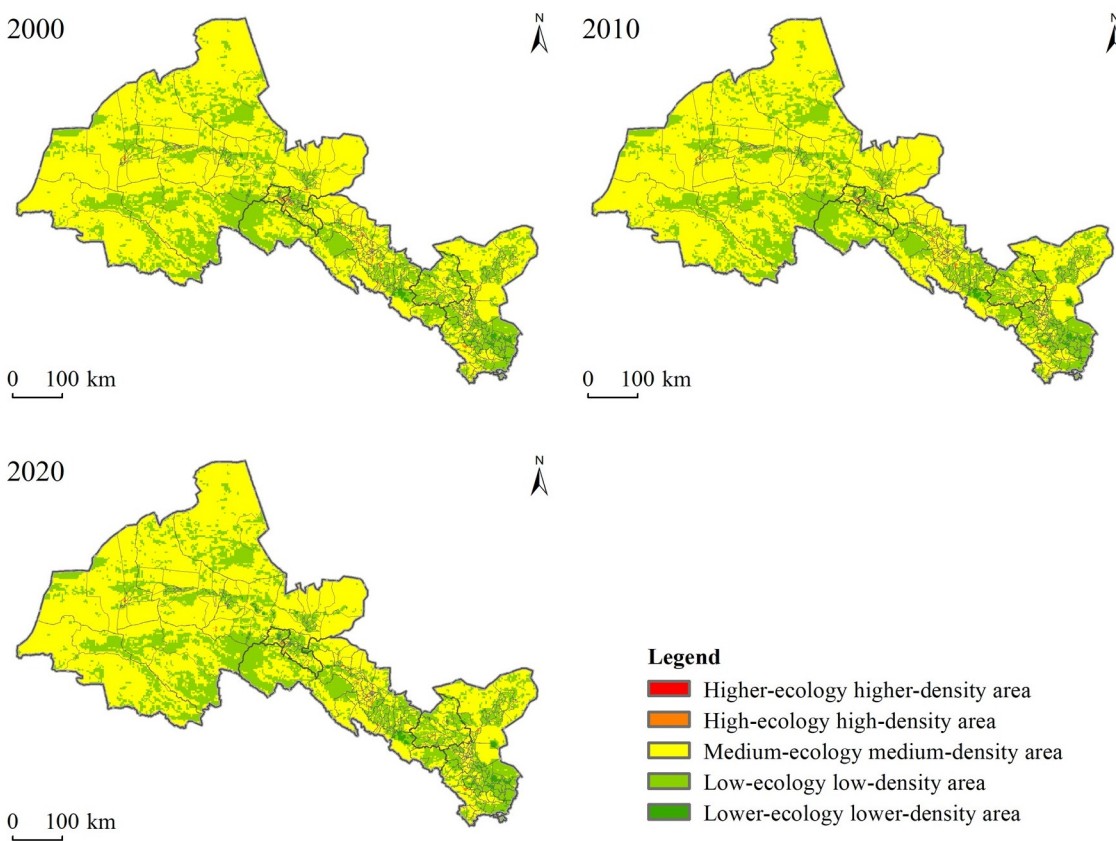

**Fig 5. Spatial distribution of the combined characteristics of rural settlement agglomeration and ecological value for the Hexi Corridor(the map was prepared in ArcGIS 10.5 using political boundaries from the Data Center for Resources and Environmental Sciences, Chinese academy of sciences (RESDC) (http://www.resdc.cn)).**

**Table 6. Characteristics of rural settlement agglomeration and ecological value in combination for the Hexi Corridor, from 2000 to 2020.**

| | Type | Higher-ecology higher-density area | High-ecology high-density area | Medium-ecology medium-density area | Low-ecology low-density area | Lower-ecology lower-density area |
|---|---|---|---|---|---|---|
| **2000** | Density (units/km$^2$) | 1.4–2.84 | 0.72–1.39 | 0.34–0.71 | 0.1–0.33 | 0–0.09 |
| | Area (km$^2$) | 7.43 | 768.48 | 165549.12 | 78010.68 | 3006.92 |
| | Percentage (%) | 0.003 | 0.31 | 66.93 | 31.54 | 1.22 |
| **2010** | Density (units/km$^2$) | 1.41–2.84 | 0.72–1.4 | 0.33–0.71 | 0.1–0.32 | 0–0.09 |
| | Area (km$^2$) | 7.43 | 698.9 | 164130.96 | 79068.95 | 3436.39 |
| | Percentage (%) | 0.003 | 0.28 | 66.36 | 31.97 | 1.39 |
| **2020** | Density (units/km$^2$) | 1.31–2.73 | 0.68–1.3 | 0.32–0.67 | 0.1–0.31 | 0–0.09 |
| | Area (km$^2$) | 12.16 | 516.33 | 163156.44 | 79989.02 | 3668.7 |
| | Percentage (%) | 0.005 | 0.21 | 65.96 | 32.34 | 1.48 |

From 2000 to 2020, it was found that the areas of higher-ecology higher-density and high-ecological high-density are nested within the oasis plain area in a patch-like manner, yet their pooled area accounts for less than 1% of the Hexi Corridor. Over time, the higher-ecology higher-density type showed a trend of increasing in overall area, while the high-ecological high-density instead decreased in area. This type was found near the inner urban part of oasis formations, where the transportation convenience is good. To be specific, it is mainly distributed in Liangzhou, Ganzhou, Jiayuguan, Suzhou, and Dunhuang, where the density of rural settlements ranged from 0.68km$^{-2}$ to 1.41km$^{-2}$, which reflects the characteristically high degree of rural settlement agglomeration in oasis areas. The medium-ecology medium-density type has the largest land coverage, accounting for 66% of the Hexi Corridor. Although highly correlated with its ecological spatial distribution, the distribution of rural settlements in this type of area is sparse and the population size is small. The topography of low-density and low-ecological type areas is relatively complex, and the traffic is inconvenient due to the surrounding desert lands and the Gobi desert, which is not conducive to the development of settlements.

## Spatial reconstruction path of rural settlements

**Accessibility level and reconstruction partition identification.** The combined characteristics of rural settlement agglomeration and ecological value reflect the coupling characteristics of spatial distribution and ecological environment of rural settlements in the Hexi Corridor. Hence, they can be used point out the direction of rural settlement reconstruction and its optimization. Considering the different land types, slope, elevation, and other spatial factors that lead to the differential influence of rural settlements upon in the study area, and according to the combination characteristics of agglomeration and ecological value, the accessibility range of higher-density higher-ecological type area was calculated in this paper. The natural break-point method was used to divide the accessibility results into five levels: grades I and IV corresponded to the highest and lowest accessibility, respectively. The grid through which the river system passes is null. Considering the spatial connection of each settlement, the boundary of rural settlement reconstruction area was preliminarily determined, and the reconstruction area was moderately optimized based on the combinations of agglomeration and ecological value. Finally, the study area was divided accordingly into four categories: urban agglomeration, central village construction, internal coordination, and ecological protection (Fig 6).

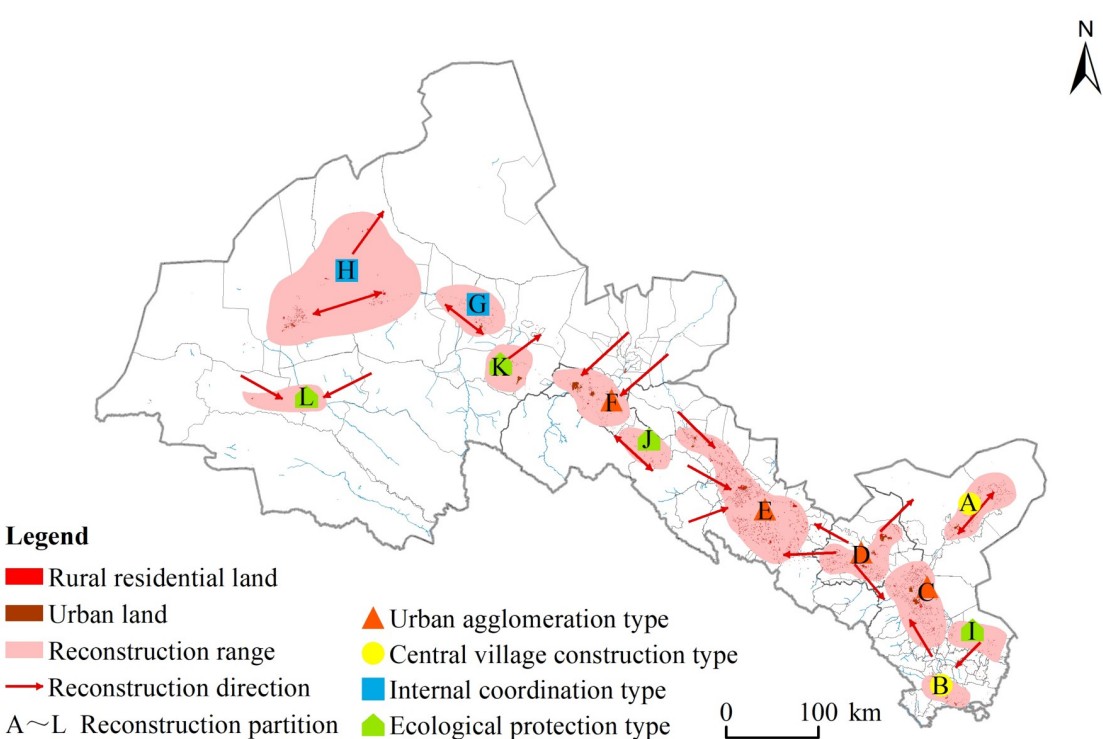

**Fig 6. Direction and partitioning of rural settlement space reconstruction in the Hexi Corridor(the map was prepared in ArcGIS 10.5 using political boundaries from the Data Center for Resources and Environmental Sciences, Chinese Academy of Sciences (RESDC) (http://www.resdc.cn)).**

These results showed that the spatial distribution and size of the reconstituted zones were closely related to the size of rural settlements, the population distribution density, and ecological value characteristics. Evidently, the urban agglomeration rural settlement type had the greatest coverage and highest accessibility, being distributed continuously in space and mainly concentrated in the plains inside the corridor. This type encompassed 189 towns in total, with a rural settlement scale of 540.89 km$^2$, or 62.82% of the total rural settlement scale. The central village construction was the rural settlement type with the second highest accessibility, being mainly distributed in 25 towns and townships of Wuwei City. The scale of its rural settlement was 73.54 km$^2$, accounting for 8.54% of the total scale of rural settlement. The internally coordinated rural settlements were mainly distributed in 38 towns and townships of Jiuquan City, and their scale 93.29 km$^2$, which is 10.84% of the total scale of rural settlements. Ecological protection type rural settlements are low-value areas of accessibility, and their spatial distribution was relatively dispersed. This type entailed 25 towns and townships overall, and the scale of its rural settlements was 28.56 km$^2$, 3.32% of the total scale of rural settlements.

**Direction recognition and path reconstruction.** According to the regional characteristics and economic development status of the different types of rural settlements, this paper puts forward reconstruction ideas and directions of four types of rural settlements. Among them, the urban agglomeration-type villages can continue to play a driving role in the radiation accompanying the Lanzhou-Xinjiang Railway, National Highway 312, Lianhuo Expressway, and other major roads in the area, so that the rural settlements in the radiation circle of the urban area would be concentrated in the cities and thereby promote the integrated

development of urban and rural areas. We encourage (1) transferring capital, population and technology to surrounding rural areas, develop rural tourism, facility agriculture and agricultural product processing industries given the local conditions; (2) implementing policies and measures to build beautiful villages; (3) addressing the sparse distribution of rural settlements along major roads; (4) enhancing the ecological and environmental quality of surrounding rural settlements; and (5) promoting the integrated development of primary, secondary, and tertiary industries in rural areas.

Central village building relies on land consolidation to settle farmers in rural consolidation areas; this approach can improve infrastructure construction, optimize the layout of the rural settlement system, and augment land use intensity by settling idle areas inside existing villages or constructing new independent areas. We suggest consolidating the position of corn, wheat and other dominant producing areas, moderately expanding the area of vegetables, fruits, high-quality special forage crops and other regional characteristic products, and developing characteristic and efficient agriculture.

The internally coordinated villages carry out the cultivation of wheat, cotton, corn and other agricultural products by guiding the adjustment of agricultural structure. We advise accelerating the renovation of medium—and low-yield farmland, raising the rate of land transfer and trusteeship, hastening the removal and relocation of aboveground attachments, and continuing to promote the construction of high-standard farmland. Other goals include vigorously developing agricultural technology; promoting the large-scale and standardized transformation of traditional breeding farms and low-level breeding communities; striving to bolster the level of large-scale cattle and sheep breeding; improving the development of rural settlements, and ensuring the basic quality of life of residents.

With the purpose of building a beautiful countryside, eco-protection-oriented villages can fully respect the will of the people, while retaining ethnic, regional, and grassland characteristics; this entails planning agricultural housing renovation projects in a scientific and rational way, steadily improving the rural living environment and not blindly building township communities, to realize ecological civilization construction. We could implement a system of banning grazing, and balancing grass and livestock to protect the ecological environment of grasslands. We should cultivate characteristic industrial clusters, promote yak breeding and other characteristic animal husbandry industries, and improve the quality of life of farmers and herdsmen alike. By relying on its own ecological advantages, rural homestay demonstration sites can be cultivated and supported with ethnic customs and characteristics, to build cultural brands with ecological rural homestay characteristics, and in process boost long-term rural revitalization and development.

## 4. Conclusions

1. There are significant differences in the scale density and spatial distribution of rural settlements in the Hexi Corridor. The overall density of rural settlements decreased from 2000 to 2020, and the spatial distribution pattern showed a significant global small-scale agglomeration trend. Spatially, the density and size of the settlements are inversely related; that is, the low-density scattered distribution lies mainly in the west, while the high-density cluster distribution is found in the east. The cold and hot spots for the degree of rural settlement agglomeration are alternately distributed in space, while the spatial agglomeration characteristics of rural settlement to urban areas and central towns are not prominent.

2. From 2000 to 2020, the ecosystem services value of the Hexi Corridor rose by 374.504 billion yuan, with a large increase occurring from 2000 to 2010. Ecological space contributed

the most to total ecosystem service value with a contribution rate of 74.93%, indicating that the overall ecological environment quality in the study area was good. The spatial differentiation of ecosystem services value in the study area is also significant, in that its spatial distribution pattern is opposite that of the degree of rural settlement agglomeration. This indicates that the spatial distribution of ecosystem services value is clearly affected by topographic factors.

3. The overall agglomeration degree and ecological value of rural settlements in the Hexi Corridor are moderate, among them, the combination type of higher-ecology higher-density is the optimal layout pattern, which is nested within the oasis plain area as patches. Accordingly, the accessibility range of the higher-density higher-ecological type area is calculated, to serve as a boundary reference, thus enabling the identification of differentiated scope area of rural settlement reconstruction. On the basis of fully considering the spatial connectivity of rural settlements, the study area could be reliably divided into four reconstruction modes: urban agglomeration type, central village construction type, internal coordination type, and ecological protection type. Finally, a reconstruction path for rural settlements is proposed for the Hexi Corridor.

## 5. Prospective

Considering that the rural settlement forms in Hexi Corridor are relatively simple and homogeneous, this paper only used the kernel density and ecological service value, and did to take other spatial factors and social and economic factors into account. For example, in the reconstruction zoning, 85.52% of the rural settlements in the study area were covered from the perspective of agglomeration. Further, the reconstruction direction of rural settlements with high dispersion was not studied, and our determination of the reconstruction direction ignored the opinions of farmers who inhabit rural settlements, rendering it relatively idealized and general reconstruction idea.

In future studies, efforts should be made in the following respects. (1) The research scale should focus further on the township unit, accurately identify the regional functioning of rural settlements, and more precisely implement the spatial reconstruction of rural settlements with the combination of multi-type settlements and multi-level objectives. (2) Investigate the spatio-temporal evolution and reconstruction strategies of rural settlements by integrating in depth spatial, socio-economic, and ecological environment factors. (3) Focus on the reconstruction and optimization of rural settlement systems, by integrating ecosystem service value vis-à-vis the prevailing ecological civilization strategy, in order to promote the harmonious development of human-land relationship in arid oasis regions.

## Author Contributions

**Conceptualization:** Xiaoying Nie.

**Data curation:** Xiaoying Nie, Wanzhuang Huang.

**Formal analysis:** Xiaoying Nie.

**Funding acquisition:** Xiaoying Nie.

**Investigation:** Xiaoying Nie.

**Methodology:** Xiaoying Nie, Chao Wang.

**Project administration:** Xiaoying Nie.

**Resources:** Xiaoying Nie.

**Software:** Xiaoying Nie, Chao Wang.

**Supervision:** Xiaoying Nie, Wanzhuang Huang.

**Validation:** Xiaoying Nie.

**Visualization:** Xiaoying Nie.

**Writing – original draft:** Xiaoying Nie.

**Writing – review & editing:** Xiaoying Nie.

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
