## [Decision Letter · Decision Letter 0]

26 Sep 2023

PONE-D-23-19951Evolution and Spatial Reconstruction of Rural Settlements with Combined Agglomeration and Ecological Value Characteristics in the Hexi Corridor, Northwest ChinaPLOS ONE

Dear Dr. nie,

Thank you for submitting your manuscript to PLOS ONE. After careful consideration, we feel that it has merit but does not fully meet PLOS ONE’s publication criteria as it currently stands. Therefore, we invite you to submit a revised version of the manuscript that addresses the points raised during the review process.

We look forward to receiving your revised manuscript.

Kind regards,

Bing Xue, Ph.D.

Academic Editor

PLOS ONE

“This research was funded by the Science and Technology Program of Gansu Province (grant number 20JR10RA296).”

4. We note that Figures 1, 3, 4, 5 and 6 in your submission contain [map/satellite] images which may be copyrighted. All PLOS content is published under the Creative Commons Attribution License (CC BY 4.0), which means that the manuscript, images, and Supporting Information files will be freely available online, and any third party is permitted to access, download, copy, distribute, and use these materials in any way, even commercially, with proper attribution. For these reasons, we cannot publish previously copyrighted maps or satellite images created using proprietary data, such as Google software (Google Maps, Street View, and Earth). For more information, see our copyright guidelines: http://journals.plos.org/plosone/s/licenses-and-copyright.

1. You may seek permission from the original copyright holder of Figures 1, 3, 4, 5 and 6 to publish the content specifically under the CC BY 4.0 license. 

Reviewers' comments:

Reviewer's Responses to Questions

**Comments to the Author**

1. Is the manuscript technically sound, and do the data support the conclusions?

Reviewer #1: Yes

Reviewer #2: Yes

2. Has the statistical analysis been performed appropriately and rigorously? 

Reviewer #1: Yes

Reviewer #2: Yes

3. Have the authors made all data underlying the findings in their manuscript fully available?

Reviewer #1: Yes

Reviewer #2: Yes

4. Is the manuscript presented in an intelligible fashion and written in standard English?

Reviewer #1: Yes

Reviewer #2: Yes

5. Review Comments to the Author

Reviewer #1: This research is really sound and informative to the international reader.

however here is some comment to more improve the paper before published

1. Improve outline of the paper for more attractive

2. the paragraph are too long and unattractive for the reader please be condense and make them more readable

3. paper needs future suggestion

4.abstract need rewrite

5. please give care for research methodology and its outline

6.Likely to research methodology, I wanna to be you give strong attention your result part. please make it more informative, readable, clear based on your data

7. Make your title more clear and informative for international knowledge searcher in this field of discipline

Reviewer #2: In this manuscript, the author applied GIS spatial analysis technology and ecosystem service value modeling to analyze the combined spatial agglomeration and ecological value characteristics of rural settlements in the Hexi Corridor, Northwest China. The study then identifies specific rural settlement reconstruction zoning and optimization directions based on rural settlement accessibility. This study is interesting and provides a related dataset.

6. PLOS authors have the option to publish the peer review history of their article (what does this mean?). If published, this will include your full peer review and any attached files.

Reviewer #1: **Yes: **Teshome Sirany

Reviewer #2: No

---

## [Author Response · Author response to Decision Letter 0]

5 Oct 2023

PONE-D-23-19951

Evolution and Spatial Reconstruction of Rural Settlements with Combined Agglomeration and Ecological Value Characteristics in the Hexi Corridor, Northwest China

PLOS ONE

To Editor:

Thank you for giving us a chance to revise the manuscript. We also thank the reviewers for their constructive suggestions to help us improve the quality of the manuscript. We have made changes in the manuscript according to the reviewers’ comments and your advice. The revised manuscript has utilized “tracked changes” to identify the revisions. A clean manuscript is also provided. We hope these changes make the manuscript acceptable for publication.

Response: 

Thanks for your comments. It has been modified according to Plos One journal template requirements.

Response: 

Thanks for your comments. All the methods in this paper need ArcGIS10.5 software to complete, and we have t permits to use the software.

“This research was funded by the Science and Technology Program of Gansu Province (grant number 20JR10RA296).”

Response: 

Thanks for your comments. Our cover letter include this amended Role of Funder statement. 

Grants and funding: This research was funded by the Science and Technology Program of Gansu Province (grant number 20JR10RA296). The funders had no role in study design, data collection and analysis, decision to publish, or preparation of the manuscript.

4. We note that Figures 1, 3, 4, 5 and 6 in your submission contain [map/satellite] images which may be copyrighted. All PLOS content is published under the Creative Commons Attribution License (CC BY 4.0), which means that the manuscript, images, and Supporting Information files will be freely available online, and any third party is permitted to access, download, copy, distribute, and use these materials in any way, even commercially, with proper attribution. For these reasons, we cannot publish previously copyrighted maps or satellite images created using proprietary data, such as Google software (Google Maps, Street View, and Earth). For more information, see our copyright guidelines: http://journals.plos.org/plosone/s/licenses-and-copyright.

1. You may seek permission from the original copyright holder of Figures 1, 3, 4, 5 and 6 to publish the content specifically under the CC BY 4.0 license. 

2. If you are unable to obtain permission from the original copyright holder to publish these figures under the CC BY 4.0 license or if the copyright holder’s requirements are incompatible with the CC BY 4.0 license, please either i) remove the figure or ii) supply a replacement figure that complies with the CC BY 4.0 license. Please check copyright information on all replacement figures and update the figure caption with source information. If applicable, please specify in the figure caption text when a figure is similar but not identical to the original image and is therefore for illustrative purposes only.c

Response: 

Thanks for your comments. The data in figures 1, 3, 4, 5 and 6 of the paper is can be obtained from the Data Center for Resources and Environmental Sciences, Chinese Academy of Sciences (RESDC) (http://www.resdc.cn); the Geospatial Data Cloud site, Computer Network Information Center, Chinese Academy of Sciences(http://www.gscloud.cn) and China’s geographic remote sensing ecological network platform (www.gisrs.cn). A data availability statement is provided in S1 File.

Response: 

Thanks for your comments. We have carefully checked the references listed in the paper to make sure that it is complete and correct.

To Reviewer:

We are very grateful for your assessment of our work. Thank you very much for your time and effort in reviewing our manuscript. We appreciate your useful comments. Please find our responses to the reviewer’s suggestions in the following section.

Reviewer #1: 

This research is really sound and informative to the international reader. However here is some comment to more improve the paper before published.

1. Improve outline of the paper for more attractive

Response: 

Thanks for your comments. We have revised the outline of the paper according to the structure and template requirements of Plos One journal.

2. the paragraph are too long and unattractive for the reader please be condense and make them more readable

Response: 

Thanks for your comments. It has been modified according to your suggestion.

3. paper needs future suggestion

Response: 

Thanks for your comments. We have added future suggestion after the conclusion of the paper.

4.abstract need rewrite

Response: 

Thanks for your comments. It has been modified according to your suggestion.

5. please give care for research methodology and its outline

Response: 

Thanks for your comments. It has been modified according to your suggestion.

6.Likely to research methodology, I wanna to be you give strong attention your result part. please make it more informative, readable, clear based on your data

Response: 

Thanks for your comments. It has been modified according to your suggestion.

7. Make your title more clear and informative for international knowledge searcher in this field of discipline

Response: 

Thanks for your comments. We have modified the title to “Evolution and Spatial Reconstruction of Rural Settlements Based on Composite Features of Agglomeration Effect and Ecological Effects in the Hexi Corridor, Northwest China”.

Reviewer #2: 

In this manuscript, the author applied GIS spatial analysis technology and ecosystem service value modeling to analyze the combined spatial agglomeration and ecological value characteristics of rural settlements in the Hexi Corridor, Northwest China. The study then identifies specific rural settlement reconstruction zoning and optimization directions based on rural settlement accessibility. This study is interesting and provides a related dataset.

Major:

1. Line 425-427: Is it feasible to determine the boundary of the reconfiguration range of rural settlements based on the accessibility measurement method, considering the spatial distribution and size of the reconstituted zones?

Response:

Thanks for your comments. The research results show that the distribution of main roads and rivers is closely related, and the reconstructed area with high grade not only maintains a certain distance from the city, but also has good contact opportunities with the outside world, and the villagers are convenient to travel, which indicates the feasibility of identifying the scope boundary of rural settlement reconstruction with the accessibility measurement method to a certain extent.

2. Line 530: Please explain the meaning of the statistic “85.52%” and provide clarification on its significance.

Response:

Thanks for your comments. 85.52 represents the proportion of the area of rural reconstruction zoning to the total area of rural settlements in the study area. As shown in Figure 1, the distribution of some rural settlements in the study area is relatively scattered, and the area of the study area is relatively large, so only the reconstruction direction of most settlements with relatively concentrated distribution is determined.

Minor:

1. Line 108-109: All numbers above 1000 should include thousands separators, such as “1,000”.

Response: 

Thanks for your comments. It has been modified according to your suggestion.

2. Line 123: Replace “Gansu Province” with “Hexi Corridor”.

Response: 

Thanks for your comments. It has been modified according to your suggestion.

3. Line 145: Add “in China” after “Statistics”.

Response: 

Thanks for your comments. It has been modified according to your suggestion.

4. Line 156: The “Ecosystem services value equivalent factor” column in Table 2 should be placed in a separate table between line 219 and line 220.

Response: 

Thanks for your comments. It has been modified according to your suggestion.

5. Line 160 and Line 258: Replace “NNI” with the correct term.

Response: 

Thanks for your comments. It has been replaced “ANN”.

6. Line 262: Consider separating the paragraph starting with “The trend surface analysis…” into separate paragraphs.

Response: 

Thanks for your comments. It has been modified according to your suggestion.

7. Line 291-298: Merge the paragraph starting with “We detected two density…” with the previous paragraph.

Response: 

Thanks for your comments. It has been modified according to your suggestion.

8. Line 313-322: Merge the paragraph starting with “Rural settlement cold spots…” with the previous paragraph.

Response: 

Thanks for your comments. It has been modified according to your suggestion.

9. Line 349: There is an error in the format “3-km × 3-km”. Please correct it.

Response: 

Thanks for your comments. It has been modified to be correct.

10. Line 392: The format “0.68·km-2 to 1.41·km-2” is incorrect. Please revise it.

Response: 

Thanks for your comments. It has been modified to be correct.

11. Line 439: Replace “(” with “,”.

Response: 

Thanks for your comments. It has been modified according to your suggestion.

12. Line 454- 460: Consider separating the segments of “Central village building…” to “efficient agriculture” into separate paragraphs.

Response: 

Thanks for your comments. It has been modified according to your suggestion.

13. Line 461-470: Consider separating the segments of “The internally coordinated…” to “residents” into separate paragraphs.

Response: 

Thanks for your comments. It has been modified according to your suggestion.

14. Line 470-481: Consider separating the segments of “With the purpose…” to “rural revitalization and development” into separate paragraphs.

Response: 

Thanks for your comments. It has been modified according to your suggestion.

15. Line 538: Replace “.” with “;” to improve clarity.

Response: 

Thanks for your comments. It has been modified according to your suggestion.

---

## [Editor Report · Decision Letter 1]

24 Oct 2023

Evolution and Spatial Reconstruction of Rural Settlements Based on Composite Features of Agglomeration Effect and Ecological Effects in the Hexi Corridor, Northwest China

PONE-D-23-19951R1

Dear Dr. nie,

We’re pleased to inform you that your manuscript has been judged scientifically suitable for publication and will be formally accepted for publication once it meets all outstanding technical requirements.

Kind regards,

Bing Xue, Ph.D.

Academic Editor

PLOS ONE
---

## [Editor Report · Acceptance letter]

30 Oct 2023

PONE-D-23-19951R1 

Evolution and Spatial Reconstruction of Rural Settlements Based on Composite Features of Agglomeration Effect and Ecological Effects in the Hexi Corridor, Northwest China 

Dear Dr. Nie:

I'm pleased to inform you that your manuscript has been deemed suitable for publication in PLOS ONE. Congratulations! Your manuscript is now with our production department. 

Kind regards, 

on behalf of

Professor Bing Xue 

Academic Editor

PLOS ONE